# Blind Remote Sensing Image Deblurring Based on Overlapped Patches’ Non-Linear Prior

**DOI:** 10.3390/s22207858

**Published:** 2022-10-16

**Authors:** Ziyu Zhang, Liangliang Zheng, Wei Xu, Tan Gao, Xiaobin Wu, Biao Yang

**Affiliations:** 1Changchun Institute of Optics, Fine Mechanics and Physics, Chinese Academy of Sciences, Changchun 130033, China; 2University of Chinese Academy of Sciences, Beijing 100049, China; 3Key Laboratory of Space-Based Dynamic & Rapid Optical Imaging Technology, Chinese Academy of Sciences, Changchun 130033, China

**Keywords:** OPNL prior, remote sensing images, overlapped patches, blind image deblurring

## Abstract

The remote sensing imaging environment is complex, in which many factors cause image blur. Thus, without prior knowledge, the restoration model established to obtain clear images can only rely on the observed blurry images. We still build the prior with extreme pixels but no longer traverse all pixels, such as the extreme channels. The features are extracted in units of patches, which are segmented from an image and partially overlap with each other. In this paper, we design a new prior, i.e., overlapped patches’ non-linear (OPNL) prior, derived from the ratio of extreme pixels affected by blurring in patches. The analysis of more than 5000 remote sensing images confirms that OPNL prior prefers clear images rather than blurry images in the restoration process. The complexity of the optimization problem is increased due to the introduction of OPNL prior, which makes it impossible to solve it directly. A related solving algorithm is established based on the projected alternating minimization (PAM) algorithm combined with the half-quadratic splitting method, the fast iterative shrinkage-thresholding algorithm (FISTA), fast Fourier transform (FFT), etc. Numerous experiments prove that this algorithm has excellent stability and effectiveness and has obtained competitive processing results in restoring remote sensing images.

## 1. Introduction

In actual imaging processes, the acquired images always face blurring problems caused by the imaging equipment and environment. In order to remove the blurs, image restoration technology has gradually attracted the attention of researchers and developed into a significant branch of image processing. The ideal image degradation process can be simply described as:(1)Y=K∗X+N,
where *Y*, *K*, *X*, and *N* represent the observed blurry image, blur kernel, original clear image, and noise, respectively, and ∗ represents the convolution operator. In fact, the cause of blurring is unknown, i.e., lack of prior information, which makes it an ill-conditioned problem with infinite solutions when simultaneously solving the blur kernel and the clear image. A traditional blind image deblurring algorithm is dedicated to finding the optimal global solution, i.e., the blur kernel, by using image information to optimize the equation, then utilizing the non-blind image deblurring algorithm to obtain a clear image.

Due to the complexity and diversity of actual situations, parametric models [1,2] obviously cannot accurately describe the blur kernel, so they are not competent for the work of image restoration. With the research results by Rudin et al. [3] in 1992, the theory of partial differential has gradually become an essential part of image restoration. Traditional image restoration models can be roughly divided into two frameworks: Maximum A Posterior (MAP) [4,5,6,7] and Variable Bayesian (VB) [8,9,10,11,12], which are based on probability theory and statistics. Considering the computation and complexity of the algorithm, although the VB framework is more stable, most methods still choose the MAP framework. At this time, the introduction of appropriate conditions can make the MAP framework avoid the problem that the naive MAP approaches with a sparse derivative prior tend to the trivial solutions, which were proven by Levin et al. [13]. The MAP framework considers that solving the blur kernel and the clear image at the same time can be regarded as solving the standard maximum posterior probability, which is expressed as:(2)$$\begin{array}{c} \left(X,K)=\arg\max P(X,K|Y)\propto \arg\max P(Y|X,K)P(X)P(K\right) \end{array}$$
where P(Y|X,K) is the noise distribution, and P(X) and P(K) are the prior distributions of the clear image and blur kernel, respectively. We take the negative logarithm of each item in Equation (Equation 2) to construct the following regularized model:(3)$$\left(X,K\right)=\arg\min\Psi \left(Y-K*X\right)+\alpha_{}\varphi \left(X\right)+\beta_{}\psi \left(K\right)$$
where Ψ(:) is the fidelity term, φ(X) and ψ(K) are the regularization functions on *X* and *K*, and α and β are the corresponding parameters.

As one of the most prominent features in images, edge information has always played an important role in image restoration algorithms. However, if there is an image that lacks sharp edges, it is necessary to take prior knowledge as the core, which can distinguish clear images from blurry images, and use image edge information as an auxiliary to achieve the desired deblurring. Channels are independent planes that store the color information of an image. The channel prior has gradually entered researchers’ vision since Pan et al. [14] verified that the distribution of dark channel pixels is significantly different for clear and blurry images. The dark channel prior performs well when dealing with natural, text, facial, and low-light images. Yan et al. [15] designed a restoration algorithm based on extreme channel prior, including dark and bright channels, to deal with insufficient dark channel pixels in the image. Ge et al. [16] proposed a non-linear channel prior with extreme channels, which has been proven by experiments to effectively solve the problem of algorithm performance degradation due to the lack of extreme pixels. At the same time, the local feature of images as the prior knowledge also shines in the field of image deblurring, e.g., local maximum gradient prior [17], local maximum difference prior [18], patch-wise minimal pixels prior [19], etc.

In this paper, inspired by the non-linear channel prior and the patch-wise minimal pixels prior, we design a new non-linear prior and develop the corresponding algorithm to deal with the task of remote sensing image restoration. To further improve operational efficiency and reduce running time, our method does not traverse all of an image’s pixels but the patches divided by the image when extracting features. Each patch will have a partial overlap to record more detailed feature information. After that, similar to how the extreme channels extract image features, the extreme pixels of each patch are found and made into two sets, i.e., the local minimum intensity set and the local maximum intensity set. We use a convex L1-norm constraint on the ratio of corresponding elements in the two local intensity sets as a non-linear prior. It is clear that OPNL prior prefers clear images rather than blurry images through the analysis of more than 5000 remote sensing images. It is difficult to solve the image restoration model based on OPNL prior directly, so we use the half-quadratic splitting method to decompose the whole model into several subproblems that are easier to solve. The non-linear term can be converted into a linear term by constructing a linear operator using the result of the previous iteration in the algorithm. The contributions of this research are as follows:(1)In this paper, we propose a new prior based on extreme pixels in patches, i.e., OPNL prior. The analysis and tests of more than 5000 remote sensing images show that OPNL prior is more conducive to clear images.(2)A new image restoration algorithm is designed based on OPNL prior for remote sensing images, which can effectively deblur while maintaining good convergence and stability.(3)The experimental results show that the proposed method outperforms the comparative methods in dealing with blurry remote sensing images. Even in the face of remote sensing images with complex textures and many details, the method can still obtain very competitive deblurring results.

The paper’s outline is as follows: Section 2 organizes the achievements made in the field of blind image deblurring over the years. In Section 3, we will elaborate on the OPNL prior. Section 4 establishes an image restoration model and designs the corresponding optimization algorithm. Section 5 presents the experimental results of our algorithms. Section 6 quantitatively analyzes the performance of our method. Section 7 is the conclusion.

## 2. Related Work

This section classifies existing blind image deblurring methods into three categories and briefly reviews recent achievements.

### 2.1. Edge Detection-Based Algorithms

Edges are favored by researchers and exist in many blind image restoration methods because they can clearly describe image features and are easy to extract. Joshi et al. [20] proposed using sharp edges to estimate blur kernel with sub-pixel accuracy to restore a clear image. Cho and Lee [21] believe that the prominent edges in an image play a dominant role in estimating blur kernel and designed algorithms, including bilateral filtering, shock filtering, and gradient magnitude thresholding to extract strong edges. Xu and Jia [22] proposed a new edge selection method to deal with strong edges that are not conducive to image restoration. By analyzing natural images and synthetic structures, Sun et al. [23] can obtain appropriate patch priors for image edges and corners and build a new patch-based image deblurring model. It is worth noting that edge detection-based algorithms fail when blurry images do not have enough suitable strong edges. However, even image prior-based algorithms still need edge information to build models, e.g., enhanced low-rank prior [24], L0-regularized intensity and gradient prior [25], and local maximum gradient prior [17].

### 2.2. Image Priors-Based Algorithms

Such algorithms typically use prior knowledge composed of image features to optimize results in favor of clear images rather than blurry images. Shan et al. [26] incorporated noise distribution and new local smoothness prior into the probability model of image deblurring to reduce artifacts. Krishnan et al. [27] concluded that L1/L2 regularization prior is more conducive to clear images after analyzing the images with Gaussian blur. Levin et al. [10] improved the conventional MAP algorithm by deriving a simple approximate MAP model without increasing the algorithm’s complexity. Kotera et al. [28] showed that the MAP model combined with heavy-tailed prior could obtain good processing results. Michaeli and Irani [29] used the deviation between the actual patches and the ideal patches in different image scales as a prior for image deblurring. Ren et al. [24] used a new enhanced low-rank prior to improve the effectiveness of deblurring, which combines low-rank priors on similar patches of blurry images and their gradient images. Zhong et al. [30] designed a high-order variational model based on the statistical characteristics of impulse noise, which can deblur and preserve image details under the interference of impulse noise. As Pan et al. [14] applied dark channels to the field of blind image deblurring with great success, sparse channels became the source of much prior knowledge. Yan et al. [15] proposed a bright channel as opposed to the dark channel and combined them into the extreme channel prior. Some improved priors based on extreme channels have also been successful, e.g., the priors of the image restoration model constructed by Yang et al. [31] and Ge et al. [16]. Zhou et al. [32] analyzed the effect of blur on different channels in the color space and established an image restoration model based on a single luminance channel prior, which is referred to as the dark channel prior. In recent years, blind image restoration algorithms that take local image features as the priors have also developed rapidly, e.g., local maximum gradient prior [17], local maximum difference prior [18], and patch-wise minimal pixels prior [19]. Our algorithm can also be viewed as such. For nighttime images, Chen et al. [33] developed a new image deblurring model by introducing the latent mapping relationship based on the saturated and unsaturated pixels of the images.

### 2.3. Deep Learning-Based Algorithms

The rapid development of deep learning technology makes it shine in various fields, and the field of image restoration is no exception. Early learning network-based algorithms still need to be designed with reference to traditional algorithms, e.g., methods proposed by Sun et al. [34] and Schuler et al. [35]. Li et al. [36] combined deep learning with traditional algorithms and used deep convolutional neural networks to realize image discrimination and feature extraction. However, this method is not suitable for dealing with highly nonuniform blurs, such as motion blur and defocus blur. Unlike algorithms relying on blur kernels to restore images, the end-to-end learning network can directly obtain clear images from blurry images through training. Nah et al. [37] creatively proposed a multi-scale convolutional neural network and used a multi-scale loss function to remove complex motion blur. Cai et al. [38] introduced channel priors into neural networks and developed the Dark and Bright Channel Prior embedded Network (DBCPeNet), which effectively handles blurring in dynamic scenes. To further improve the algorithm’s computational efficiency and deblurring quality, Zhang et al. [39] and Suin et al. [40] both developed a patch-based hierarchical network. Pan et al. [41] expected the algorithm to have more functions and restore images from more aspects. Based on physical models, they trained end-to-end using a generative adversarial network framework that finally solves problems such as image deblurring, dehazing, and deraining. Many algorithms focus on dealing with a certain kind of image or blur, e.g., ID-Net [42], DCTResNet [43], and LSFNet [44]. However, due to many parameters, deep learning-based algorithms require long-term and large-scale training to obtain excellent processing results.

## 3. Overlapped Patches’ Non-Linear Prior

This section introduces the OPNL prior and demonstrates that it is beneficial for clear images rather than blurry images.

### 3.1. The Overlapped Patches’ Extreme Intensity Pixels

OPNL prior is based on sets of extreme pixels with overlapping patches. The extreme pixels in an overlapping patch *i* are defined as:(4)$$Mi\left(X\right)\left(i\right)=\min_{\left(x,y\right)\in \Omega_{1}\left(i\right)}\left(\min_{c\in \left\{r,g,b\}\right.}X\left(x,y,c\right)\right)$$
(5)$$Ma\left(X\right)\left(i\right)=\max_{\left(x,y\right)\in \Omega_{2}\left(i\right)}\left(\max_{c\in \left\{r,g,b\}\right.}X\left(x,y,c\right)\right)$$
where (x,y) are pixel coordinates, $\Omega_{1}$ and $\Omega_{2}$ are the domain of patch pixels, and *c* represents a color channel. Since conventional remote sensing images only have one channel, Equations (4) and (5) can be described as:(6)$$Mi\left(X\right)\left(i\right)=\min_{\left(x,y\right)\in \Omega_{1}\left(i\right)}X\left(x,y\right)$$
(7)$$Ma\left(X\right)\left(i\right)=\max_{\left(x,y\right)\in \Omega_{2}\left(i\right)}X\left(x,y\right)$$

Due to the presence of blur in an image, as demonstrated by Pan et al. and Yan et al., extreme pixels in a patch are averaged with surrounding pixels, which results in Mi increasing and Ma decreasing for each patch. The inferences are as follows:1.Let Mi(B) and Mi(C) represent the local minimum intensity pixel sets of a clear image and the corresponding blurry image, respectively, then:
(8)Mi(B)(i)≥Mi(C)(i)2.Let Ma(B) and Ma(C) represent the local maximum intensity pixel sets of a clear image and the corresponding blurry image, respectively, then:
(9)Ma(B)(i)≤Ma(C)(i)

### 3.2. The Overlapped Patches’ Non-Linear Prior

Inspired by Ge et al. [16], in order to enhance the discrimination of image features for clear images and blurry images, we use the extreme intensity pixels of overlapping patches to form a non-linear term OPNL:(10)$$N\left(X\right)\left(i\right)=\frac{\min_{\left(x,y\right)\in \Omega_{1}\left(i\right)}X\left(x,y\right)}{\max_{\left(x,y\right)\in \Omega_{2}\left(i\right)}X\left(x,y\right)}=\frac{Mi(X)(i)}{Ma(X)(i)}\;$$

The dimensions of $\Omega_{1}$ and $\Omega_{2}$ in the prior are the same. Furthermore, if there is a patch such that Ma(X)(l)=0 and Mi(X)(l)=0 hold, then N(X)(l)=0.

For the case where N(X)(i) of other patches is not 0, it can be deduced that:(11)$$1\leq \frac{Mi(B)(i)}{Mi(C)(i)}\leq \frac{Mi(B)(i)Ma(C)(i)}{Mi(C)(i)Ma(B)(i)}=\frac{N(B)(i)}{N(C)(i)}$$

Accordingly, we can design a new prior, i.e., $\psi \left(X\right)={∥N(X)∥}_{1}$, accumulated by the convex L1-norm. It can be seen from the above formula: N(B)(i)≥N(C)(i), i.e., ${∥N(B)∥}_{1}>{∥N(C)∥}_{1}$, in which one case is excluded that ${∥N(B)∥}_{1}={∥N(C)∥}_{1}$, which requires all elements in image *X* to be equal. In experiments to test the effectiveness of OPNL prior, we selected 5200 images from the AID dataset [45] and added one of the eight blur kernels from the Levin dataset [13] to each image. As shown in Figure 1a, the large pixel values of OPNL for blurry images are significantly more than those for clear images. Figure 1b shows that minimizing the OPNL prior is more conducive to obtaining a clear image in the image deblurring model.

## 4. Solving Process of Image Deblurring Algorithm

Under the MAP framework, we introduce a new prior, i.e., OPNL prior, into the image deblurring model and develop a corresponding solution algorithm. The objective function of our algorithm is:(12)$$\min_{K,X}{∥K⊗X-Y∥}_{2}^{2}+\alpha {∥N(X)∥}_{1}+\beta {∥\nabla X∥}_{0}+\gamma {∥K∥}_{2}^{2}$$
where α, β, and γ are the weights of the corresponding regularization terms. The first term is the fidelity term, which aims to mitigate the influence of noise on the restoration result by minimizing it. The other items are the correlation term of OPNL prior, the regularization term of the image gradient, and the constraint term of the blur kernel constrained by the L2-norm to smooth the blur kernel. It is not practical to directly solve the objective function. Thus, we can use the projected alternating minimization (PAM) method to decompose the model to obtain the clear image and blur kernel, respectively. We replace Ma(X) of the OPNL prior term with $Ma(X_{P})$, where $X_{P}$ is the image result of the previous iteration:(13)$$\min_{X}{∥K⊗X-Y∥}_{2}^{2}+\alpha {∥\frac{Mi(X)}{Ma(X_{p})}∥}_{1}+\beta {∥\nabla X∥}_{0}$$
(14)$$\min_{K}{∥K⊗X-Y∥}_{2}^{2}+\gamma {∥K∥}_{2}^{2}$$

Based on the multi-scale image pyramid, the estimated blur kernel can be obtained after the loop iteration. Then, with the blur kernel and the blurry image as initial conditions, an image non-blind deblurring algorithm is used to restore the clear image.

### 4.1. Estimating the Latent Image

To further reduce the difficulty of solving, we apply the half-quadratic splitting method to separate the prior term and the regularization term of the gradient by introducing auxiliary variables. Equation (Equation 13) can be rewritten as:(15)$$\begin{array}{c} \min_{X,t,r}{∥K⊗X-Y∥}_{2}^{2}+\alpha {∥\frac{t}{Ma(X_{p})}∥}_{1}+\beta {∥r∥}_{0}\\ +\lambda_{1}{∥Mi(X)-t∥}_{2}^{2}+\lambda_{2}{∥\nabla X-r∥}_{2}^{2} \end{array}$$
where $\lambda_{1}$ and $\lambda_{2}$ are the penalty parameters, and *t* and *r* are auxiliary variables. When $\lambda_{1}$ and $\lambda_{2}$ grow to infinity, Equations (13) and (15) are equivalent. After that, other variables are fixed, and each variable can be solved alternately. Based on the blurry image, the auxiliary variable *t* is solved by:(16)$$\min_{t}\alpha {∥\frac{t}{Ma(X_{p})}∥}_{1}+\lambda_{1}{∥Mi(X)-t∥}_{2}^{2}$$

The relationship between the non-linear operators (Mi(X) and Ma(X)) and the pixels of the original image is expressed by constructing the mapping matrices (*I* and *A*), that is:(17)$$I\left(i,j^{*}\right)=\left\{\begin{array}{c} 1,\\ 0, \end{array}\begin{array}{c} j^{*}=\arg\min_{j\in \Omega_{1}\left(i\right)}X\left(j\right)\\ otherwise \end{array}\right.$$
(18)$$A\left(i,j^{*}\right)=\left\{\begin{array}{c} 1,\\ 0, \end{array}\begin{array}{c} j^{*}=\arg\max_{j\in \Omega_{2}\left(i\right)}X\left(j\right)\\ otherwise \end{array}\right.$$

The same as in Ge et al. [16], the sparse matrix I by explicit calculation, Equation (Equation 16) is rewritten as follows:(19)$$\min_{t}\alpha {∥\frac{t}{\mathrm{Ma}(X_{p})}∥}_{1}+\lambda_{1}{∥\mathrm{IX}-t∥}_{2}^{2}$$
where t, X, and Ma(x) are vector forms of *t*, *X*, and *Ma*(*x*), respectively. Further, we rewrite Equation (Equation 19):(20)$$\min_{e}\alpha {∥e∥}_{1}++\lambda_{1}{∥\mathrm{IX}-diag(\mathrm{Ma}(X))e∥}_{2}^{2}$$

We can handle the classic convex L1-regularized problem by applying the fast iterative shrinkage-thresholding algorithm (FISTA) [46], whose contraction operator is:(21)$$D_{c}{\left(x\right)}_{i}=\mathrm{sgn}\left(x_{i}\right)\max(|x_{i}-c\left|,0\right)$$

The solution process of *t* is shown in Algorithm 1.
**Algorithm 1** Solving auxiliary variables t in (20).**Input**: $A=diag(\mathrm{Ma}\left(X_{p}\right))$, B=IX, αα, $\lambda_{1}$,$g=\max(eig\left(A^{T}A\right))$, m=1, $q_{1}=1$.Maximum iteration *M*, initial value $y_{1}=s_{0}$.**While**m≤M     $s_{m}=T_{\alpha /\lambda_{1}}\left(y_{m}-gA^{T}\left({\mathrm{Ay}}_{m}-B\right)\right)$     $q_{m+1}=\frac{1+\sqrt{1+4q_{m}^{2}}}{2}$     $y_{m+1}=s_{m}+\frac{q_{m}-1}{q_{m+1}}\left(s_{m}-s_{m-1}\right)$     m=m+1**End while**$t={\mathrm{As}}_{M}$

Then solve the auxiliary variable *r*:(22)$$\min_{r}\beta {∥r∥}_{0}+\lambda_{2}{∥\nabla X-r∥}_{2}^{2}$$

The result of Equation (Equation 22) is:(23)$$r=\left\{\begin{array}{c} 0,\\ \nabla X, \end{array}\right.\begin{array}{c} {\left|\nabla X\right|}^{2}<\beta /\lambda_{2}\\ otherwise \end{array}$$

After completing the solution of the auxiliary variables, we solve clear image *X* by:(24)$$\min_{X}{∥K⊗X-Y∥}_{2}^{2}+\lambda_{1}{∥Mi(X)-t∥}_{2}^{2}+\lambda_{2}{∥\nabla X-r∥}_{2}^{2}$$

We replace all items with matrix vector form:(25)$$\min_{X}{∥\mathrm{KX}-Y∥}_{2}^{2}+\lambda_{1}{∥\mathrm{IX}-t∥}_{2}^{2}+\lambda_{2}{∥\nabla X-r∥}_{2}^{2}$$K represents the blur kernel Toeplitz form. Equation (Equation 25) appears to be a simple least squares problem, but it cannot be solved directly using Fast Fourier Transform (FFT). Therefore, we continue to introduce an auxiliary variable u:(26)$$\begin{array}{c} \min_{X}{∥\mathrm{KX}-Y∥}_{2}^{2}+\lambda_{1}{∥\mathrm{Iu}-t∥}_{2}^{2}+\lambda_{2}{∥\nabla X-r∥}_{2}^{2}+\lambda_{3}{∥X-u∥}_{2}^{2} \end{array}$$
where $\lambda_{3}$ is a penalty parameter. We decompose Equation (Equation 26) into two sub-problems to solve the clear image X and the auxiliary variable u, respectively:(27)$$\min_{X}\lambda_{1}{∥\mathrm{Iu}-t∥}_{2}^{2}+\lambda_{3}{∥X-u∥}_{2}^{2}$$
(28)$$\min_{X}{∥\mathrm{KX}-Y∥}_{2}^{2}+\lambda_{2}{∥\nabla X-r∥}_{2}^{2}+\lambda_{3}{∥X-u∥}_{2}^{2}$$

Both sub-problems have closed solutions. The solution of Equation (Equation 27) is:(29)$$u=\frac{\lambda_{1}I^{T}t+\lambda_{3}X}{\lambda_{1}I^{T}I+\lambda_{3}}$$

Equation (Equation 28) is solved by FFT:(30)$$X=F^{-1}\left(\frac{\bar{F(K)}F\left(Y\right)+\lambda_{2}\bar{F(\nabla )}F\left(r\right)+\lambda_{3}F\left(u\right)}{\bar{F(K)}F\left(K\right)+\bar{F(\nabla )}F\left(\nabla \right)+\lambda_{3}}\right)$$
where *F*, $F^{-1}\left(\cdot \right)$, and $\bar{F(\cdot )}$ represent FFT, the inverse FFT, and the complex conjugate operators of FFT, respectively. The above solution process is shown in Algorithm 2.
**Algorithm 2** Solving the Latent Image *X*.**Input**: Blurry image *Y* and blur kernel *K*.Initialize $\lambda_{1}$, X←Y.**For**i← 1 to 5, **do**     Solve *t* using Algorithm 1.     Initialize $\lambda_{3}$.   **For** j← 1 to 4 **do**     Solve u using Equation (Equation 27).     Initialize $\lambda_{2}$.     **While** $\lambda_{2}<\lambda_{2}^{\max}$.      Solve r using Equation (Equation 22).      Solve X using Equation (Equation 28).      $\lambda_{2}\leftarrow 2\lambda_{2}$.     **End while**     $\lambda_{3}\leftarrow 4\lambda_{3}$.   **End for**   $\lambda_{1}\leftarrow 4\lambda_{1}$.**End for****Output**: latent image X.

### 4.2. Estimating the Blur Kernel

As proposed in the literature [14,15,16,17,18,19,25,31,32], a more accurate blur kernel can be estimated by replacing the image gray term in Equation (Equation 14) with the image gradient term. Equation (Equation 14) can be modified as follows:(31)$$\min_{K}{∥K⊗\nabla X-\nabla Y∥}_{2}^{2}+\gamma {∥K∥}_{2}^{2}$$

FFT can estimate the blur kernel:(32)$$K=F^{-1}\left(\frac{\bar{F(\nabla X)}F\left(\nabla Y\right)}{\bar{F(\nabla X)}F\left(\nabla X\right)+\gamma }\right)$$

Note that the blur kernel *K* requires non-negative and normalization processing. The solution process is shown in Algorithm 3.
**Algorithm 3** Estimating the blur kernel *K*.**Input**: Blurry image *Y*.Initialize *K* with results from the coarser level.**While**i≤max_iter, **do**     Solve for *X* using Algorithm 2.     Solve for *K* using Equation (Equation 31).**End while****Output**: Blur kernel *K* and intermediate latent image *X*.

### 4.3. Details about the Algorithm

This section describes the settings for the parameters in our algorithm. The algorithm uses a coarse-to-fine image pyramid with a down-sampling factor of $\sqrt{2}/2$, and the number of loops for each layer is 5. In the loop of each layer, the algorithm will complete the image estimation, blur kernel estimation, and normalization, in turn. Then, the blur kernel estimated in the loop of this layer is expanded by up-sampling and passed to the next layer. The algorithm solution process is shown in Figure 2. Through a large number of experiments, we usually set the relevant parameters as α=0.002−0.01, β=0.002−0.004, and γ=2, the patch_size is 20×20, and the patch overlap rate is f=25%. The maximum number of loops in Algorithm 1 is set to 500. All parameters are not unique and can be adjusted according to needs. Finally, using the blurry image and the estimated blur kernel as the initial conditions, a clear image will be obtained by the non-blind image deblurring algorithm.

## 5. Experiment Results

Remote sensing interpretation experts selected a total of 30 scenarios, including 10,000 remote sensing images, from Google Maps. These images constitute a large-scale aerial image dataset, i.e., the AID dataset [45]. All experimental tests are conducted on the AID dataset. This section shows the processing ability of our algorithm for remote sensing images by comparing it with four different prior algorithms, i.e., dark channel (Dark) prior [14], L0 regular intensity and gradient (L0) prior [25], patch-wise minimal pixels (PMP) prior [19], and non-linear channel (NLC) prior [16].

### 5.1. Simulated Remote Sensing Image Experiment

First, the ability of our algorithm to remove the blur is tested, which is often encountered in the process of remote sensing imaging. We selected four images of high quality from the AID dataset, as shown in Figure 3, in which motion blur, Gaussian blur, and defocus blur were added for testing. Three representative and full-reference evaluation indicators are selected, i.e., Peak-Signal-to-Noise Ratio (PSNR), Structural-Similarity (SSIM) [47], and Root Mean Square Error (RMSE).

#### 5.1.1. Motion Blur

We set the relevant parameters of the motion blur, i.e., the angle is $0^{\circ }$, and the displacement is 10 pixels. Table 1 shows the evaluation results of the images with motion blur processed by all algorithms. For motion blur, the L0 method performs poorly. It can be seen that using the L0-norm to constrain the pixels and gradients in the image will cause severe over-sharpening. The dark channel prior is very effective for image processing with few texture details, but there will still be slight residual over-sharpening when processing other images. Although PMP, NLC, and our method have better visual effects in most processing results, our method has achieved higher objective evaluation indicators. For Figure 3b, with very complex texture, other methods fail to smooth the tiny details that result in the restored image with irregular edges and poor evaluation of objective indicators. Figure 4 shows the restoration results of some images.

#### 5.1.2. Gaussian Blur

We set the relevant parameters of the Gaussian blur, i.e., the size is 20 × 20, and the standard deviation is 0.5. Table 2 shows the evaluation results of images with Gaussian blur processed by all algorithms. When faced with the interference of Gaussian blur, the images processed by Dark, L0, and PMP all have over-sharpening that becomes more and more serious as the complexity of the textures increases, i.e., Figure 3b cannot be restored. The NLC algorithm generally performs well, i.e., most of the processing results have good visual effects. However, it retains too many tiny details when dealing with Figure 3b, which leads to a deterioration in image quality. By comparison, the images restored by our algorithm have advantages in both visual effects and objective evaluation indicators. Figure 5 shows the restoration results of some images.

#### 5.1.3. Defocus Blur

We set the relevant parameters of the defocus blur, i.e., the radius is 2. Table 3 shows the evaluation results of images with defocus blur processed by all algorithms. For defocus blur, Dark and L0 cannot recover more image details, and the processing results of these two also have different degrees of over-sharpening. The performance of the other three methods is basically good. Still, for Figure 3b with a complex texture, only NLC and our method can achieve satisfactory results for visual effect, but our algorithm is also slightly better in objective evaluation indicators. Comparing the results processed by these methods, it can be seen that our algorithm can competently deal with the effect of defocus blur on image quality. Figure 6 shows the restoration results of some images.

### 5.2. Real Remote Sensing Image Experiment

Finally, we observe the ability of our algorithm to solve real problems. A total of five images were used for testing, which consisted of four blurry images selected from the AID dataset and a target image obtained in the experiment, as shown in Figure 7. Due to the lack of original reference images, the full-reference evaluation indicators will be replaced by no-reference evaluation indicators, i.e., Entropy (E) [48], Average Gradient (AG), and Point sharpness (P) [49]. Table 4 shows the evaluation results of real remote sensing. Although Dark and L0 have obtained higher objective evaluation indicators, the processing results of the two methods have serious over-sharpening, which causes the image edges to be unsmooth and the details to look very messy. The performance of PMP is excellent for remote sensing images with less detail. However, when more tiny details are in the image, the restoration results will leave artifacts and generate false information. Compared with NLC, our algorithm achieves the same competitive subjective visual effect and slightly outperforms the objective evaluation indicators. The comprehensive evaluation shows that our algorithm has a good ability to solve practical problems. Figure 8, Figure 9, Figure 10, Figure 11 and Figure 12 show the restoration results.

## 6. Analysis and Discussion

This section presents the performance analysis of our proposed algorithm, including the effectiveness of OPNL prior, the influence of hyper-parameters, algorithm convergence, computational speed, and algorithm limitations. All the tests are based on the Levin dataset [13], consisting of four images and eight blur kernels. To maintain test accuracy, we uniformly specify the number of iterations and estimated blur kernel size and use the same non-blind restoration method for all algorithms. Quantitative evaluation parameters are chosen as Error-Ratio [13], Peak-Signal-to-Noise Ratio (PSNR), Structural-Similarity (SSIM) [47], and Kernel Similarity [50]. All experiments are run on a computer with an Intel Core i5-1035G1 CPU and 8 GB RAM.

### 6.1. Effectiveness of the OPNL Prior

Theoretically, the OPNL prior tends to clear images in the minimization problem, i.e., the restoration model based on OPNL prior can complete the task of image deblurring. However, the performance of OPNL prior in practice still needs to be quantitatively evaluated. Figure 13 shows the results of comparing our algorithm with other algorithms, Dark, L0, PMP, and NLC. It can be seen that the cumulative Error-Ratio of the other methods, except L0, all have little difference, but our algorithm has higher PSNR and SSIM. In summary, OPNL prior has proven to be effective in restoring degraded images in theory and practice.

### 6.2. Effect of Hyper-Parameters

The proposed restoration model mainly includes five super parameters, i.e., α, β, γ, patch_size, and overlap rate (*f*). To explore the impact of changes in hyper-parameters on the processing results, we adopt a single-variable method, i.e., change only one parameter at a time, and calculate the kernel similarity between the estimated blur kernel and the ground truth kernel. The experimental results are shown in Figure 14. A large number of experiments show that the algorithm is stable, whose processing results will not be affected by significant changes in the hyper-parameters.

### 6.3. Algorithm Convergence and Running Time

The projected alternating minimization (PAM) algorithm aims to find the optimal solution for the image restoration model. Reference [6] considers that the delay normalization of the blur kernel in the iterative process of the PAM algorithm makes the algorithms, which are based on the total variation, converge. Compared with reference [6], OPNL prior and L0-norm of the image gradient in our algorithm undoubtedly increase the complexity of the model. Using the PAM algorithm and the half-quadratic splitting method can simplify the restoration model into several sub-problems, all of which have convergent solutions. However, the convergence of the restoration model still needs to be further verified. Based on the Levin dataset, the convergence can be quantitatively tested by calculating the mean value of the objective function with Equation (Equation 12) and the mean value of kernel similarity on the premise of several iterations at the optimal scale of the image pyramid. From the results in Figure 15, it can be seen that our algorithm converges after 20 iterations, and the kernel similarity tends to be stable after 25 iterations, both of which prove the effectiveness of our method.

In addition, we also test the running time of each algorithm, which is shown in Table 5. Our algorithm can obtain more competitive results in less time by comprehensive comparison.

### 6.4. Algorithm Limitations

Although our algorithm has excellent performance, it still has limitations. First, our algorithm cannot take into account the effects of blur and noise simultaneously. The deblurring ability of our algorithm will decrease when the image is seriously polluted by noise, such as stripe noise caused by the non-uniform response of the detector. It means that image restoration requires additional steps to denoise. In addition, the proposed algorithm builds an image pyramid, which uses the PAM algorithm, the half-quadratic splitting method, and others in the loop of each layer. This structure will undoubtedly increase the algorithm’s computational complexity and running time, which is a common problem with traditional algorithms. Therefore, future research will focus on designing algorithms with broader applications and faster operation based on OPNL prior to create conditions for practical engineering applications.

## 7. Conclusions

We reduce the algorithm’s computational complexity and running time by changing the way of extracting image features from traversing all pixels to patches segmented from the image that partially overlap each other. According to the extreme pixels extracted from each patch, these are designed with overlapped patches’ non-linear prior, which has been proven to favor clear images in the energy minimization problem, and the corresponding image deblurring algorithm. A large number of comparative experiments have confirmed that the restoration results of remote sensing images obtained by this algorithm are better than other algorithms. Even in the face of remote sensing images with complex texture details, our algorithm can still restore satisfactory images. It is believed that the proposed algorithm can further promote the research of remote sensing image restoration technology. 

## Figures and Tables

**Figure 1 sensors-22-07858-f001:**
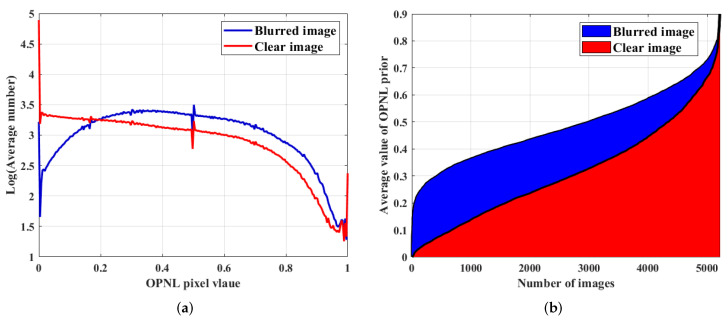
(**a**) The average pixel value distribution of OPNL for 5200 images. (**b**) The average value of OPNL prior for 5200 images.

**Figure 2 sensors-22-07858-f002:**
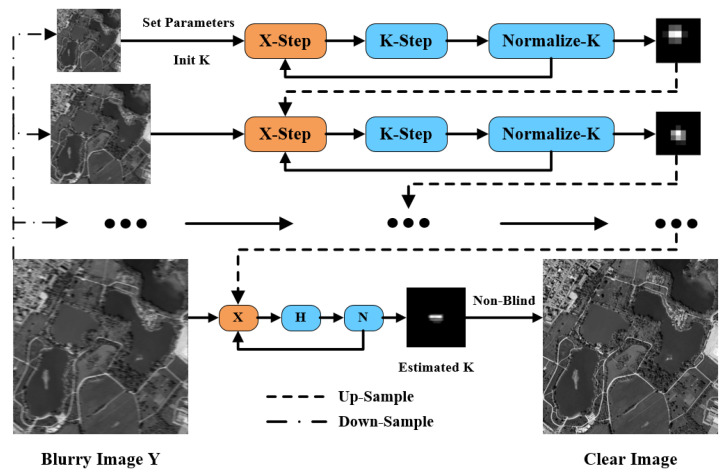
A brief flow chart of our algorithm.

**Figure 3 sensors-22-07858-f003:**
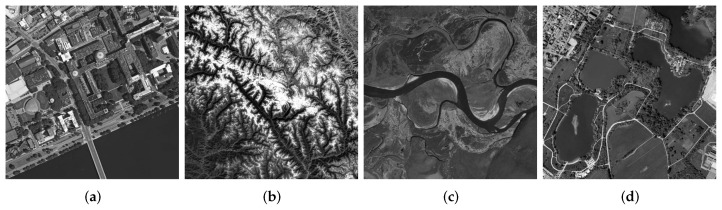
Selected remote sensing image. (Simulate). (**a**) School, (**b**) Mountain, (**c**) River, (**d**) Park.

**Figure 4 sensors-22-07858-f004:**
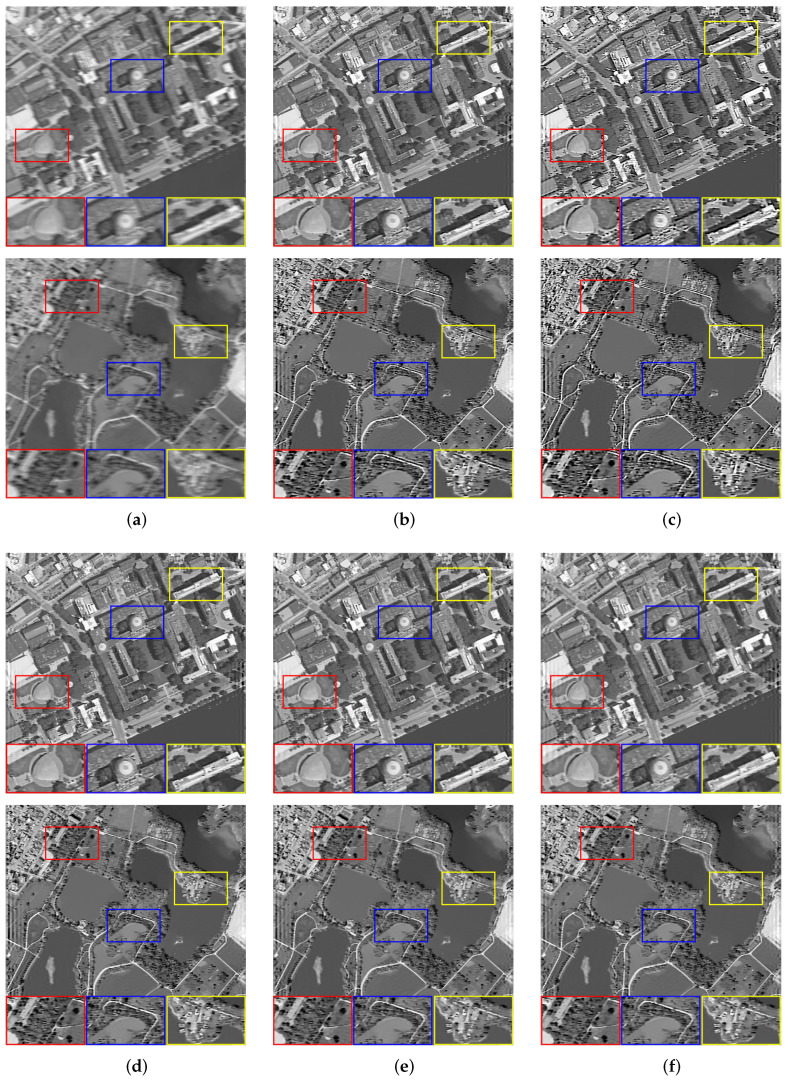
The restoration results of different methods for Figure 3a,d with motion blur. (**a**) Blurry Image, (**b**) Dark [14], (**c**) L0 [25], (**d**) PMP [19], (**e**) NLC [16], (**f**) Ours.

**Figure 5 sensors-22-07858-f005:**
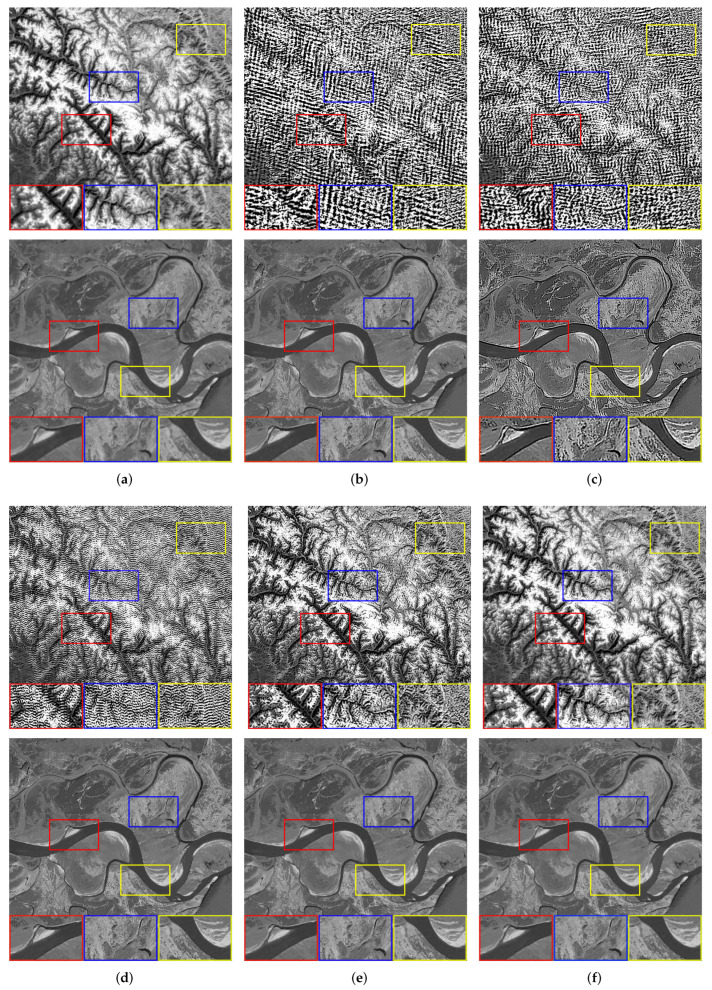
The restoration results of different methods for Figure 3b,c with Gaussian blur. (**a**) Blurry Image, (**b**) Dark [14], (**c**) L0 [25], (**d**) PMP [19], (**e**) NLC [16], (**f**) Ours.

**Figure 6 sensors-22-07858-f006:**
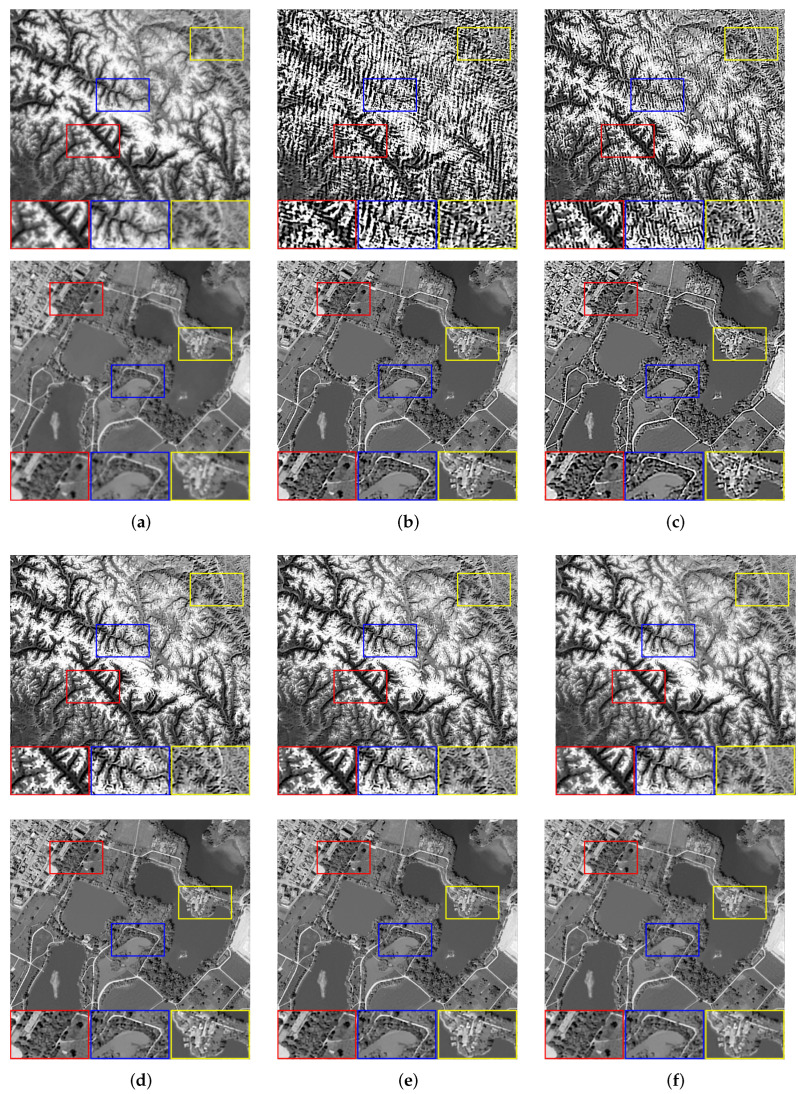
The restoration results of different methods for Figure 3b,d with defocus blur. (**a**) Blurry Image, (**b**) Dark [14],(**c**) L0 [25], (**d**) PMP [19], (**e**) NLC [16], (**f**) Ours.

**Figure 7 sensors-22-07858-f007:**
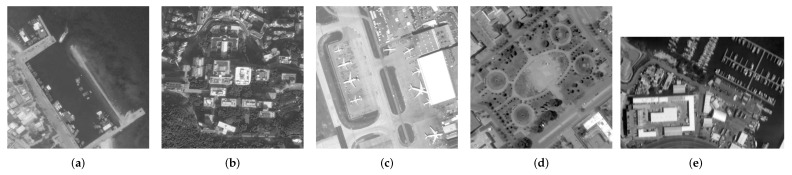
Selected remote sensing image. (Real). (**a**) Port, (**b**) School∗, (**c**) Airport, (**d**) Square, (**e**) Target Image.

**Figure 8 sensors-22-07858-f008:**
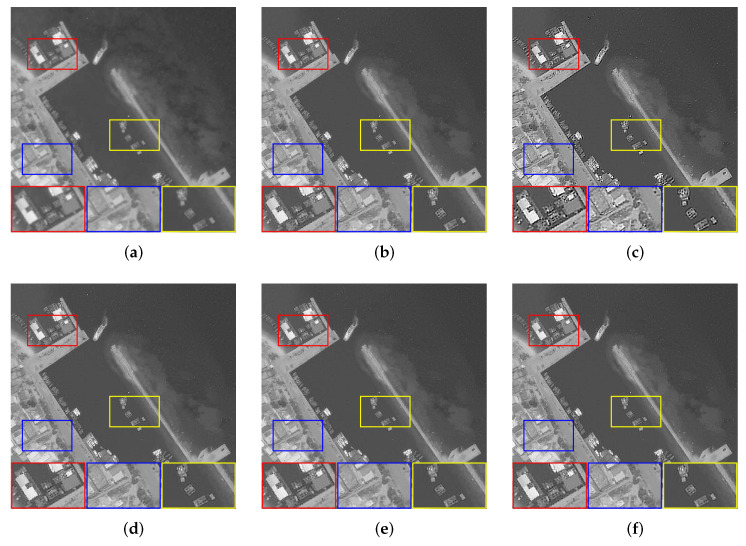
The restoration results of different methods for Figure 7a. (**a**) Blurry Image, (**b**) Dark [14], (**c**) L0 citeref25, (**d**) PMP [19], (**e**) NLC [16], (**f**) Ours.

**Figure 9 sensors-22-07858-f009:**
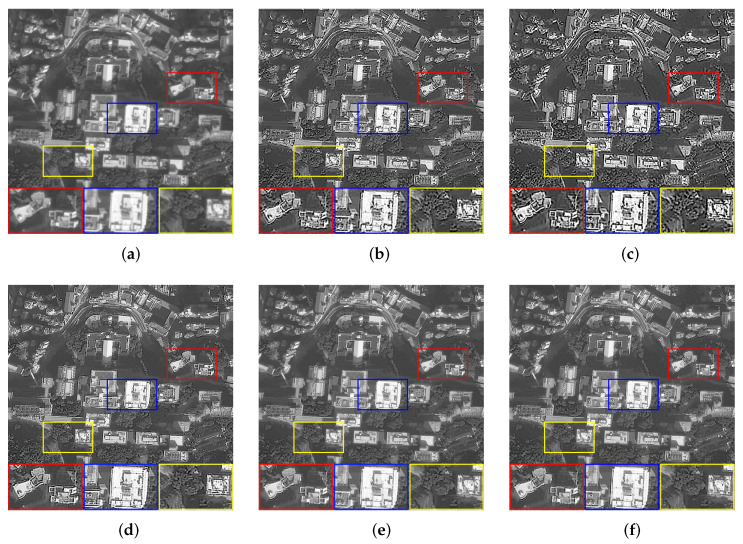
The restoration results of different methods for Figure 7b. (**a**) Blurry Image, (**b**) Dark [14], (**c**) L0 [25], (**d**) PMP [19], (**e**) NLC [16], (**f**) Ours.

**Figure 10 sensors-22-07858-f010:**
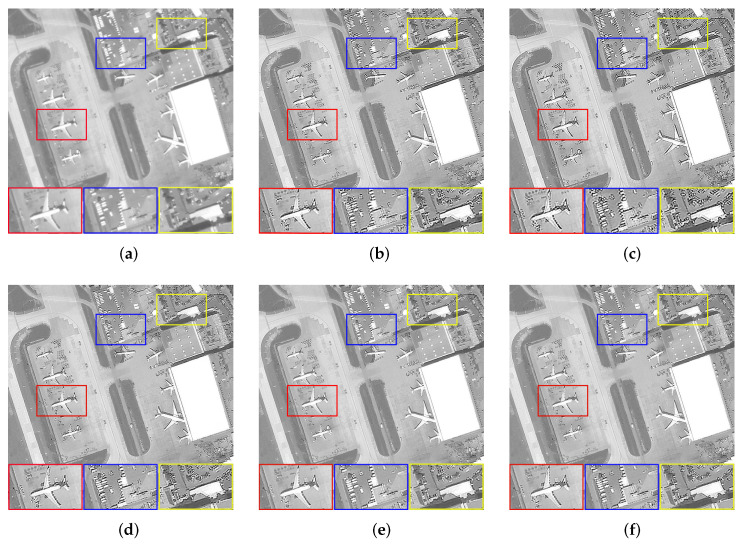
The restoration results of different methods for Figure 7c. (**a**) Blurry Image, (**b**) Dark [14], (**c**) L0 [25], (**d**) PMP [19], (**e**) NLC [16], (**f**) Ours.

**Figure 11 sensors-22-07858-f011:**
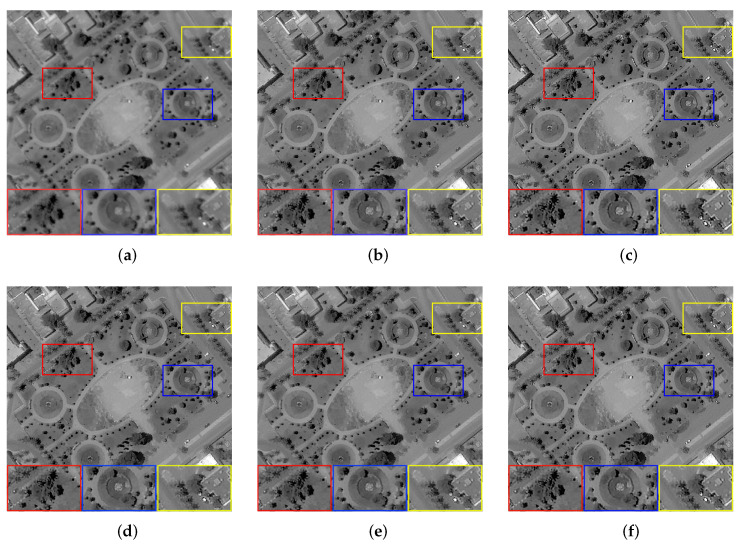
The restoration results of different methods for Figure 7d. (**a**) Blurry Image, (**b**) Dark [14], (**c**) L0 [25], (**d**) PMP [19], (**e**) NLC [16], (**f**) Ours.

**Figure 12 sensors-22-07858-f012:**
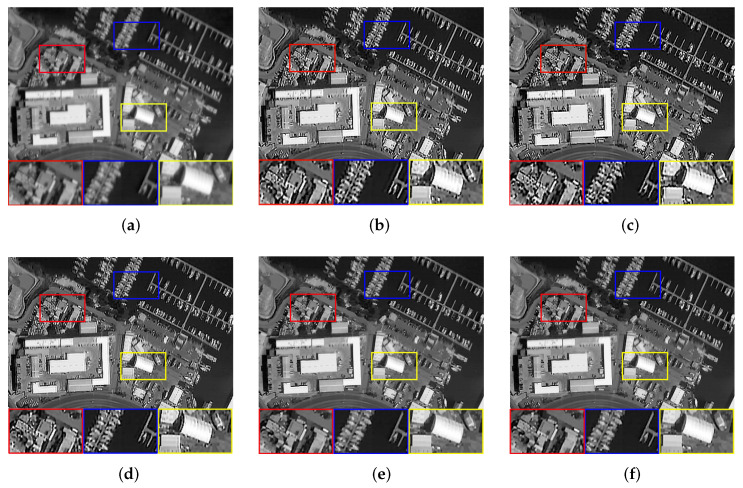
The restoration results of different methods for Figure 7e. (**a**) Blurry Image, (**b**) Dark [14], (**c**) L0 [25], (**d**) PMP [19], (**e**) NLC [16], (**f**) Ours.

**Figure 13 sensors-22-07858-f013:**
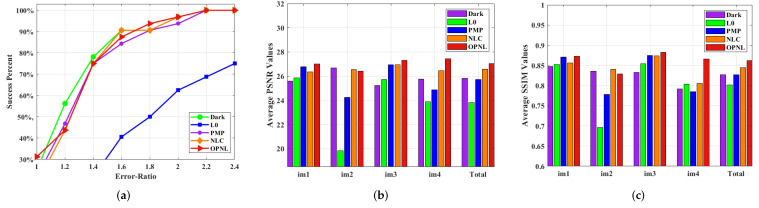
Quantitative evaluation on the benchmark dataset [13]. (**a**) Comparisons in terms of cumulative Error-Ratio. (**b**) Comparisons in terms of average PSNR. (**c**) Comparisons in terms of average SSIM.

**Figure 14 sensors-22-07858-f014:**
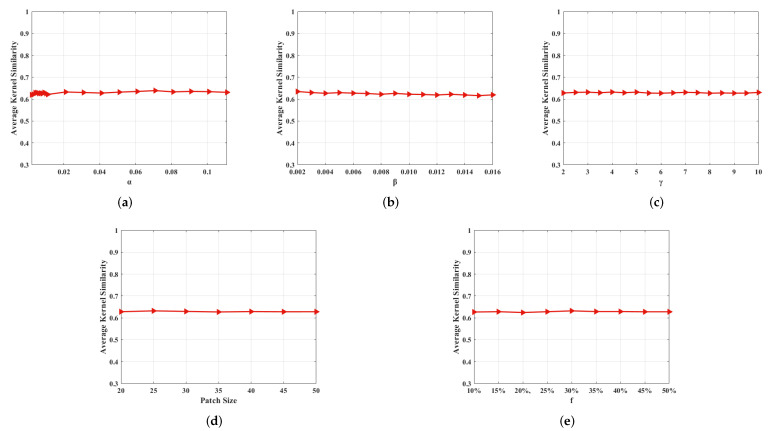
Sensitivity analysis of the hyper-parameters in our method. (**a**) Effect of α on kernel similarity. (**b**) Effect of β on kernel similarity. (**c**) Effect of γ on kernel similarity. (**d**) Effect of patch_size on kernel similarity. (**e**) Effect of *f* on kernel similarity.

**Figure 15 sensors-22-07858-f015:**
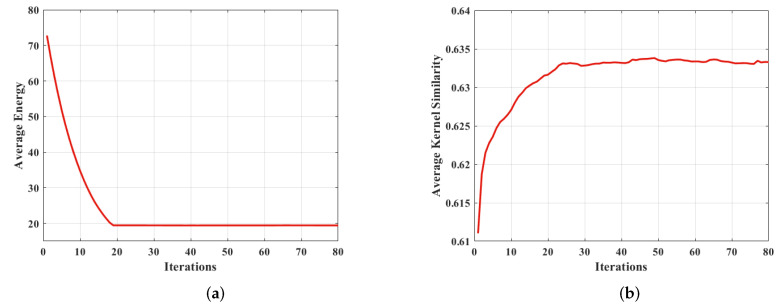
Convergence analysis of the proposed method. (**a**) The average value of the objective function (12) under the optimal scale of the image pyramid. (**b**) Kernel Similarity.

**Table 1 sensors-22-07858-t001:** Objective Evaluation Results of Remote Sensing Images with Motion Blur.

Method	Figure 4a			Figure 4b		
	PSNR	SSIM	RMSE	PSNR	SSIM	RMSE
**Dark** [14]	24.9247	0.8343	$2.08\times 10^{-4}$	8.7337	0.1263	$5.94\times 10^{-5}$
$L_{0}$ [25]	17.389	0.6035	$2.81\times 10^{-4}$	9.6171	0.2032	$7.51\times 10^{-5}$
**PMP** [19]	25.5164	0.8466	$1.07\times 10^{-4}$	10.4565	0.3279	$1.78\times 10^{-4}$
**NLC** [16]	26.0395	0.8514	$1.55\times 10^{-4}$	10.8196	0.3499	$2.67\times 10^{-4}$
**Ours**	**27.0279**	**0.8594**	$9.21\times 10^{-5}$	**17.0709**	**0.5676**	$6.28\times 10^{-5}$
**Method**	**Figure 4c**			**Figure 4d**		
	**PSNR**	**SSIM**	**RMSE**	**PSNR**	**SSIM**	**RMSE**
**Dark** [14]	29.9602	0.8143	$2.14\times 10^{-4}$	17.9493	0.6659	$2.19\times 10^{-4}$
$L_{0}$ [25]	25.1494	0.7386	$2.43\times 10^{-4}$	14.139	0.534	$2.52\times 10^{-4}$
**PMP** [19]	29.9402	0.8139	$2.31\times 10^{-4}$	22.5968	0.7977	$2.64\times 10^{-4}$
**NLC** [16]	28.9047	0.7755	$2.45\times 10^{-4}$	23.4888	0.8028	$1.91\times 10^{-4}$
**Ours**	**31.0716**	**0.8327**	$1.90\times 10^{-4}$	**24.4108**	**0.83**	$1.82\times 10^{-4}$

**Table 2 sensors-22-07858-t002:** Objective Evaluation Results of Remote Sensing Images with Gaussian Blur.

Method	Figure 4a			Figure 4b		
	PSNR	SSIM	RMSE	PSNR	SSIM	RMSE
**Dark** [14]	12.0795	0.4805	$3.46\times 10^{-4}$	1.2252	0.0336	$4.41\times 10^{-4}$
$L_{0}$ [25]	10.3751	0.4253	$4.18\times 10^{-4}$	1.1397	0.0072	$4.24\times 10^{-4}$
**PMP** [19]	13.9406	0.5948	$3.19\times 10^{-4}$	4.3068	0.065	$3.41\times 10^{-5}$
**NLC** [16]	21.32	**0.8159**	$3.88\times 10^{-4}$	11.8677	0.4549	$4.76\times 10^{-4}$
**Ours**	**22.741**	0.8074	$2.17\times 10^{-4}$	**17.7199**	**0.5875**	$1.16\times 10^{-4}$
**Method**	**Figure 4c**			**Figure 4d**		
	**PSNR**	**SSIM**	**RMSE**	**PSNR**	**SSIM**	**RMSE**
**Dark** [14]	24.927	0.6602	$1.55\times 10^{-4}$	11.4001	0.4046	$1.18\times 10^{-4}$
$L_{0}$ [25]	14.5766	0.2655	$2.11\times 10^{-4}$	7.5502	0.3127	$4.96\times 10^{-5}$
**PMP** [19]	26.9558	0.7792	$1.61\times 10^{-4}$	14.1703	0.5774	$1.08\times 10^{-4}$
**NLC** [16]	27.4396	0.7738	$1.45\times 10^{-4}$	20.1692	0.7744	$3.40\times 10^{-5}$
**Ours**	**31.132**	**0.8943**	$1.43\times 10^{-4}$	**23.8839**	**0.8481**	$4.37\times 10^{-6}$

**Table 3 sensors-22-07858-t003:** Objective Evaluation Results of Remote Sensing Images with Defocus Blur.

Method	Figure 4a			Figure 4b		
	PSNR	SSIM	RMSE	PSNR	SSIM	RMSE
**Dark** [14]	25.7106	0.868	$9.54\times 10^{-5}$	6.528	0.0915	$1.55\times 10^{-4}$
$L_{0}$ [25]	20.6962	0.759	$6.79\times 10^{-5}$	10.413	0.3589	$2.42\times 10^{-4}$
**PMP** [19]	26.0404	0.8709	$6.82\times 10^{-5}$	18.9793	0.7026	$1.78\times 10^{-4}$
**NLC** [16]	27.2596	0.892	$2.45\times 10^{-5}$	21.3035	0.7968	$1.29\times 10^{-4}$
**Ours**	**28.7608**	**0.909**	$1.87\times 10^{-5}$	**24.8868**	**0.8684**	$3.96\times 10^{-5}$
**Method**	**Figure 4c**			**Figure 4d**		
	**PSNR**	**SSIM**	**RMSE**	**PSNR**	**SSIM**	**RMSE**
**Dark** [14]	31.7588	0.8616	$1.60\times 10^{-4}$	21.7453	0.781	$8.82\times 10^{-5}$
$L_{0}$ [25]	30.6986	0.8476	$1.82\times 10^{-4}$	16.2103	0.6318	$6.96\times 10^{-5}$
**PMP** [19]	32.1846	0.8671	$1.83\times 10^{-4}$	27.5899	0.8956	$4.23\times 10^{-5}$
**NLC** [16]	30.2685	0.8337	$1.39\times 10^{-4}$	27.5728	0.8918	$9.69\times 10^{-5}$
**Ours**	**32.478**	**0.869**	$1.35\times 10^{-4}$	**28.2989**	**0.8983**	$3.00\times 10^{-5}$

**Table 4 sensors-22-07858-t004:** Objective Evaluation Results on Real Remote Sensing Images.

Method	Figure 7a			Figure 7b			Figure 7c			Figure 7d			Figure 7e		
	E	AG	P	E	AG	P	E	AG	P	E	AG	P	E	AG	P
**Dark** [14]	6.4757	0.0181	0.1263	6.9836	0.0968	0.6781	7.2132	0.0925	0.6286	6.9755	0.0267	0.1833	7.1581	0.0976	0.6845
$L_{0}$ [25]	6.5899	0.0316	0.2196	6.9261	0.1269	0.8879	7.2365	0.1072	0.7283	6.9977	0.0328	0.2266	7.1004	0.1023	0.7172
**PMP** [19]	6.4734	0.0182	0.1252	6.9269	0.0752	0.5251	7.2233	0.0852	0.5765	6.9745	0.0268	0.1843	7.1725	0.0845	0.5918
**NLC** [16]	6.4623	0.0168	0.1156	6.8072	0.0459	0.3182	7.2153	0.0587	0.3963	6.9682	0.0249	0.1715	7.2145	0.0408	0.2828
**Ours**	6.4826	0.0189	0.1301	6.8313	0.0509	0.3528	7.2225	0.0681	0.4592	6.9798	0.029	0.199	7.2279	0.0434	0.3008

**Table 5 sensors-22-07858-t005:** Running Time (in seconds) Comparison.

Method	125 × 125	255 × 255	600 × 600
**Dark** [14]	53.3	166.83	790.27
$L_{0}$ [25]	12.92	31.61	146.96
**PMP** [19]	18.59	19.79	48.19
**NLC** [16]	25.33	74.47	475.18
**Ours**	20.08	51.07	371.61

## Data Availability

The AID data used in this paper are available at the following link: http://www.captain-whu.com/project/AID/ (accessed on 27 April 2022). The Levin data used in this paper are available at the following link: www.wisdom.weizmann.ac.il/~levina/papers/LevinEtalCVPR09Data.zip (accessed on 9 May 2022).

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
