# Peer review of "Blind Remote Sensing Image Deblurring Based on Overlapped Patches’ Non-Linear Prior"

_sensors, 2022, doi:10.3390/s22207858_

Round 1
Reviewer 1 Report
The authors present a method for remote images deblurring. Their introduction is well written defining in detail the problem. Their literature review covers all the current research approaches, while their methodology is described clear and in detail. Their experiments are well designed. Their conclusions are supported by their experimental results.
They should increase the size of figures 4-8 because the results are not displayed clearly.
Also in Table 5 the reference for the NLC method is missing
Author Response
Thank you so much for your suggestions for my manuscript, we've made the corrections. Please see the attachment.

Reviewer 2 Report
Ziyu Zhang's ORCID identifier is broken. All authors should have valid ORCID identifiers with minimum information.
The use of the English language must be completely revised. Please notice that a mere checking of typos and grammar is not enough. There are several grammatically correct expressions that convey ambiguous or wrong meaning, for instance, in "Due to the complexity and diversity of actual situation, the parametric model [1,2]" it seems there is a single parametric model (because of "the parametric model," rather than "parametric models"). This is wrong. The authors must request the assistance of professional proofreading with good knowledge in the field. Another example, to the best of this reviewer's knowledge, there is no "partial differential theory;" maybe the authors refer to the use of partial differential equations. Also related to this sentence, this reviewer does not agree that the research has shifted to that field; please make an updated review of the literature, and you will find plenty of current references about deblurring based on models.
Punctuation must also be carefully revised.
The second line of (2) should not be proportional, but equal to the first line.
Eq. (3) is not a "regular model," but a regularised model.
Avoid the use of qualifiers as, for instance, "to achieve an excellent deblurring effect." In this case, prefer "to achieve the desired deblurring."
Explain, in words, what is a "channel prior" before discussing its use, variations, and applications.
Is the expression "multiple patches divided by the image" correct? I do not think so. The authors seem to refer to an image divided in patches.
In Section 3 we find for the first time that the authors use the Red, Green, and Blue channels as input data. This is in conflict with the article title, which refers to Remote Sensing images. Remote Sensing images often have more channels, spanning from multispectral to hyperspectral data (not to mention polarimetric data). This limitation should be clear from the beginning (including the title).
What is the meaning of "accumulated by the convex L1-norm"?
Provide references to the AID and Levin datasets.
"I", in the sentence before (19), should be in bold italic font. Please check all the paper for notation consistency.
The images in the experimental results are too small. The authors should choose fewer, and present them enlarged.
What is the angle in Section 5.1.1? What is the defocus blur in Section 5.1.3?
In Section 5.2 the authors use the Entropy as a blind measure of quality, and they refer to the article by Shannon. This article handles communication data (time series), and it is not clear how the authors used the Entropy. Please clarify.
The authors may consider using the metric M proposed by
Gomez, L. / Ospina, R. / Frery, A. C., Unassisted Quantitative Evaluation of Despeckling Filters. Remote Sensing, 2017, Vol. 9, No. 9
which, although proposed for speckled images, can be applied to each channel. Notice that, as discussed in that article, the M metric can also be used to tune the parameters.
The first sentence of Section 6.1 is not clear. Please rephrase.
Author Response

(The authors gave the same response as above.)

Reviewer 3 Report
This paper developed a new prior, called OPNL, for remote sensing image deblurring. To solve the resulting model, the authors develop an effective algorithm based on PAM. Experimental results on real data sets demonstrate the effectiveness of the proposed model. Overall, the paper is well written and can be easily understood. According to this reviewer, the paper can be accepted after the following minor corrections:
1. The authors need to proofread the paper thoroughly as there are a few typos. For instance, Eq. (23) and “?” in Table 5.
2. Missing reference: below Section 3.2 “Inspired by Ge et al.”
Author Response

(The authors gave the same response as above.)

Round 2
Reviewer 2 Report
The manuscript still has stylistic problems that prevent further technical analysis. As a token, the second sentence from the Abstract reads "Thus, it becomes the method to obtain a clear image by establishing a restoration model only 2 based on the blurry image under the premise that the cause of blur is unknown." which is not comprehensible.
I will gladly review a full reviewed version. The authors must either hire a professional reviewer, or use Grammarly.
Author Response
Thank you so much for your suggestions for my manuscript. We have modified the sentences that are difficult to understand and used Grammarly to correct the grammatical problems in the article comprehensively. Please see the attachment.
